# Modelling NHS England 111 demand for primary care services: a discrete event simulation

Richard Pilbery ,[1] Madeleine Smith,[2] Jonathan Green,[3] Daniel Chalk ,[4] Colin A O'Keeffe[5]

¹Research and Development, Yorkshire Ambulance Service NHS Trust, Wakefield, UK
²Business Intelligence, NHS Devon, Exeter, UK
³Faculty of Health, University of Plymouth, Plymouth, UK
⁴Medical School, University of Exeter, Exeter, UK
⁵School of Health and Related Research, The University of Sheffield, Sheffield, UK

**Correspondence to**
Richard Pilbery;
r.pilbery@nhs.net

## ABSTRACT

**Objectives** This feasibility study aimed to model in silico the current healthcare system for patients triaged to a primary care disposition following a call to National Health Service (NHS) 111 and determine the effect of reconfiguring the healthcare system to ensure a timely primary care service contact.

**Design** Discrete event simulation.

**Setting** Single English NHS 111 call centre in Yorkshire.

**Participants** Callers registered with a Bradford general practitioner who contacted the NHS 111 service in 2021 and were triaged to a primary care disposition.

**Primary and secondary outcome measures** Face validity of conceptual model. Comparison between real and simulated data for quarterly counts (and 95% CIs) for patient contact with emergency ambulance (999), 111, and primary and secondary care services. Mean difference and 95% CIs in healthcare system usage between simulations and difference in mean proportion of avoidable admissions for callers who presented to an emergency department (ED).

**Results** The simulation of the current system estimated that there would be 39 283 (95% CI 39 237 to 39 328) primary care contacts, 2042 (95% CI 2032 to 2051) 999 calls and 1120 (95% CI 1114 to 1127) avoidable ED attendances. Modifying the model to ensure a timely primary care response resulted in a mean percentage increase of 196.1% (95% CI 192.2% to 199.9%) in primary care contacts, and a mean percentage decrease of 78.0% (95% CI 69.8% to 86.2%) in 999 calls and 88.1% (95% CI 81.7% to 94.5%) in ED attendances. Avoidable ED attendances reduced by a mean of −26 (95% CI −35 to −17).

**Conclusion** In this simulated study, ensuring timely contact with a primary care service would lead to a significant reduction in 999 and 111 calls, and ED attendances (although not avoidable ED attendance). However, this is likely to be impractical given the need to almost double current primary care service provision. Further economic and qualitative research is needed to determine whether this intervention would be cost-effective and acceptable to both patients and primary care clinicians.

## STRENGTHS AND LIMITATIONS OF THIS STUDY

⇒ This study benefits from utilisation of a robust method (discrete event simulation) that has been widely used and accepted as a method to simulate healthcare system changes.
⇒ The model was informed by relevant stakeholders and a linked dataset.
⇒ The model and its source code has been made available free and open source for others to interrogate and develop.
⇒ It is a simple working model and does not take account of factors that have previously been associated with inappropriate emergency department attendance, such as age, sex, ethnicity and clinical input into the 111 call.
⇒ The data that informed the model only covers a discrete region in England, which may affect generalisability.

## INTRODUCTION

The National Health Service (NHS) 111 service aims to assist members of the public with urgent medical care needs and is the successor to the NHS Direct service in England. Following pilots in four sites it was rolled out nationally, with the final site going live in England in 2014, and in 2019/2020 received over 19 million calls.[1] Its key founding objective was to provide easy access to support for the public with urgent care needs, to ensure they received the 'right care, from the right person, in the right place, at the right time'.[2] It is also the key component of the 24/7 Integrated Urgent Care Service outlined in the NHS Long Term Plan.[3]

The triage outcome of approximately 55% of calls made to the English NHS 111 call service, is a referral to a primary care service. The triage process includes the method of contact (eg, callers may be asked to contact a primary care service themselves, or a referral made by 111 to a primary care service to contact the patient directly), and a time frame within which this should occur. If a timely service (defined as a primary care contact within the time specified as part of the triage outcome) cannot be provided to patients, it

is possible that this will result in patients calling 999 or attending emergency departments (EDs) directly. We have previously reported that just under half (47.6%) of callers to 111 triaged to a primary care service disposition made contact with a primary care service as their first healthcare interaction.[4] There was evidence that patients more frequently attended an ED when they had not had previous contact with a primary care service. However, rates of avoidable attendance were similar whether callers had contact with a primary care service prior to attendance at ED or not.

Other studies examining caller behaviour and the impacts on the wider healthcare system[5 6] have used datasets from 2017 and earlier, and patterns of activity identified then, may not be applicable now.

This study aimed to develop a discrete event simulation (DES) to model the current healthcare system for callers triaged to a primary care disposition. DES models allow for the simulation of queuing problems, in which entities (such as patients) queue for processes that require resources.[7] Our primary objective was to develop a model which represented the current healthcare system in relation to callers to 111 who received a primary care service disposition following triage. Our secondary objective was to use the model to estimate the predicted impact and associated resourcing required to ensure a timely primary care contact for all callers to the 111 service in a region of Yorkshire.

## METHODS
### Conceptual modelling
Prior to developing a simulation that represents the trajectories of patients within the model, it is necessary to develop a conceptual model: an appropriately simplified collection of assumptions about the components and structure of the 'real-world'[8] that can act as a blueprint for the in silico model development. This technique has its origins in soft systems methodology,[9] and enabled us to work with stakeholders in the ambulance service, NHS 111, general practice and EDs to coproduce processes and capture their knowledge of the health and social care system. This coproduction is not only crucial in terms of improving the accuracy of the model, but also an important component of stakeholder engagement and the formation of strategic partnerships.[10]

### Data
In tandem with the conceptual modelling phase, we obtained routine, retrospective data from the Connected Yorkshire research database, which provides linked patient-level data for approximately 1.2 million citizens across the Bradford and Airedale region of Yorkshire. Datasets include 111 and emergency ambulance (999) call data, as well as primary and secondary care (including ED and in-patient activity). Data submitted to Connected Yorkshire contain a unique identifier in the form of a pseudonymised NHS number (used throughout the NHS as a unique identifier for patient care). In this study, we used it, along with the date and time of access to services, to track a caller's healthcare journey.

We identified all 111 calls between 1 January 2021 and 31 December 2021 for patients who were triaged to a primary care disposition and registered with a general practitioner (family physician, GP) in the Bradford area at the time of the call. Subsequent healthcare system access in the 72 hours following the first (index) call, was identified by programmatically searching for the patient's unique identifier in the 111, 999 calls, primary care, hospital ED and in-patient admission datasets.[4]

Triage to a primary care service in this study refers to a range of services, including referral to the patient's own GP, or an out-of-hours GP service, pharmacy, community, optician, mental health or maternity service. Depending on the triage disposition and service availability, referrals are either made directly by 111 with a request for a service to contact the caller, or the caller is advised to contact/attend the service directly. In the source data, over 96% of primary care disposition referrals were to a GP (or out-of-hours service) and of the other services, only pharmacies, mental health and optician service interactions would not appear in the data. However, these services only comprised 2% of triage outcomes in the source data.[4]

### Discrete event simulation
Using[11] taxonomy, we determined that DES was a good choice of operational research methodology to address the research question. DES is a flexible modelling technique that can represent complex behaviour of callers to 111, and their interactions with the healthcare system simultaneously. DES can also incorporate capacity and resource constraints and account for chance, for example, the stochastic behaviour of callers.[7 12] DES is also useful, because it makes it possible to ask 'what if' questions such as the research question. It has been used widely in healthcare, examining healthcare systems operations, and disease progression, screening and health behaviour modelling.[7 13]

We created an in silico version of the conceptual model using the programming language Python (V.3.9.6) and the SymPy library (V.4.0.1[14]). In addition, we created a web-based dashboard in order to present the results of the simulation and provide an opportunity for users/stakeholders to run their own simulations. To achieve this, we used Plotly Dash (V.5.11.0[15]) and created a fully contained environment in docker (V.4.2.0.7078) which is available from the study GitHub repository (https://github.com/RichardPilbery/MOOOD-study).

Patients enter the model at the point they are triaged to a primary care disposition by 111 and have exited the service. The 'arrival' rate of callers into the model is determined by sampling from an exponential probability distribution with lambda set to the 1/mean arrival rate for 111 calls, stratified by hour, yearly quarter and whether the call occurred at a weekend. When a patient is created in the simulation, they are assigned a primary care disposition

**Main simulation creates callers and exits when 1 year's worth of calls has been generated**

**Caller simulation runs within main simulation to manage healthcare trajectory for an individual caller**

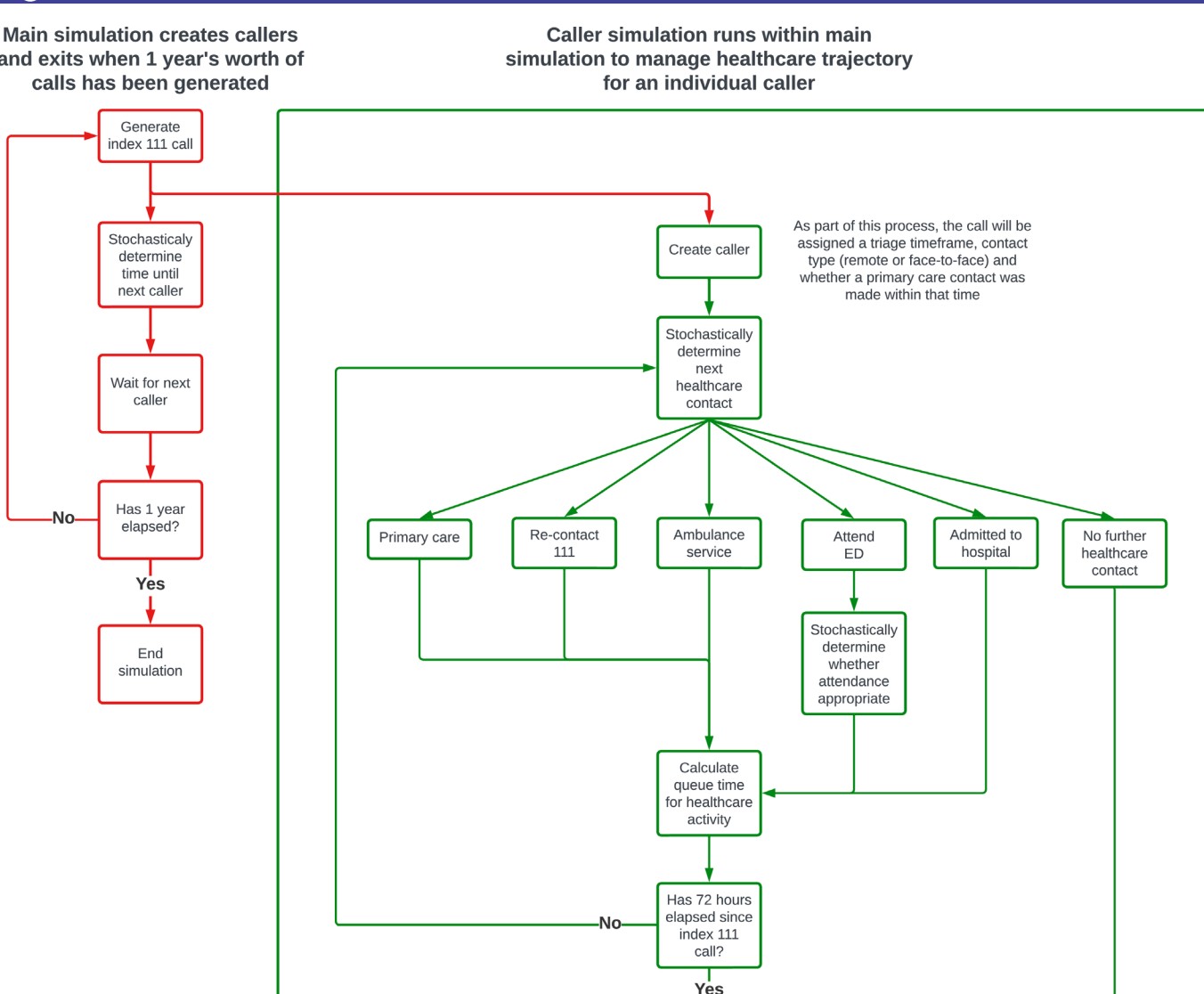

**Figure 1** Programmatic simulation flow chart. ED, emergency department.

with an associated triage acuity (figure 1). This consists of a time frame within which a contact with a primary care service should be made, and specifies whether the contact should be face to face or remotely, for example by telephone. The allocation is random, but weighted according to the proportions seen in the Connected Yorkshire data. Once the patient has been allocated a primary care disposition, the uniform distribution is sampled to determine whether the primary care contact will be achieved in the specified triage time frame. This is assumed to be the case when the sampled value is less than proportion calculated from the Connected Yorkshire dataset.

Patient trajectory through the healthcare system is determined using transition probabilities calculated from the Connected Yorkshire dataset using the statistics software, R.[16] The next healthcare service (including no further

contact) is determined by random sampling, weighted by whether a primary care service contact was made immediately after the index 111 call, and the yearly quarter. If the next healthcare service access is determined to be the ED, then the uniform distribution is sampled to determine whether the admission will be avoidable. As before, this is assumed to be the case when the sampled value is less than proportion calculated from the Connected Yorkshire dataset. The proportions are stratified by primary care disposition and whether initial contact was made with a primary care service.

For this study, an ED attendance is classed as being 'avoidable' when it meets the O'Keeffe *et al*[17] criteria. They defined an avoidable attendance as a patient presenting to a consultant-led ED which provides a 24-hour service with full resuscitation facilities and designated accommodation

for the reception of emergency care patients (referred to as a type 1 ED[18]), but who do not receive investigations, treatments or referral that required the facilities of a type 1 ED.

Unless the patient is allocated to 'no further healthcare contact', in which case the patient exits the simulation, the 'queue' time that is, the time the patient waits until the service is accessed and the time that the healthcare activity takes, is calculated. The 'queue' time is calculated by sampling from a distribution specific to the current healthcare service being access and the next, for example, index 111 call to ED attendance. Activity times are calculated by sampling from distributions specific to the service being accessed. In both cases, the optimal distribution was chosen by the Python library fitter (V.1.5.2[19]) based on the distribution with the lowest sum of squared errors.

Patients remain in the simulation for a maximum of 72 hours unless they are allocated to 'no further healthcare contact' prior to this. The model generates 111 calls for a period of 1 year of simulated time.

## Analysis

Once the model was developed, we performed 100 runs of 1 year's worth of simulated 111 calls and monitored the patient's trajectory over the subsequent 72 hours. As the model is stochastic, multiple runs of the model were necessary to ensure that results are representative and not due to stochastic noise. A summary of all simulations was reported as the mean and 95% CIs for counts and proportions.

Since all models are approximations of the 'real world', we used the historic data provided by the Connected Yorkshire dataset to see how closely our model matches with real data (the primary objective), visually and descriptively analysing the difference in temporal distribution of calls between the actual and simulated data, as well as comparing quarterly aggregated healthcare service access by patients, following the index 111 call. Finally, we performed a visual assessment of patient trajectory, to determine whether it approximated actual patient behaviour.

For the secondary objective, we undertook the 'what if' analysis, examining the hypothetical situation whereby all index 111 calls with a primary care disposition received a timely (ie, within the specified call triage time) response from a primary care service. No capacity constraints were placed on primary care services in order to estimate the true resourcing that would be required to meet the demand appropriately. We calculated the mean difference and 95% CIs in healthcare system usage between simulations and determined the difference in mean proportion of avoidable admissions for callers who presented to an ED.

## Patient and public involvement

The application and protocol for this study was reviewed by the Yorkshire Ambulance Service NHS Trust patient research ambassador. In addition, Connected Bradford has an active patient and public involvement group and a representative attended the approvals board for this study.

## RESULTS

We visually compared outputs from the model with historic patient data and observed sufficient similarity in the distribution pattern of calls across the day, which gave us confidence that the model was sufficiently capturing the variability within the real-world data (figures 2 and 3). A 111 call activity in both simulations was similar to the actual call activity seen in the real data, although overall, the simulation did slightly overestimate the number of calls, particularly between 00:00 and 06:00 hours (figure 2A). The simulation tended to underestimate 999 call activity and overestimate all other healthcare service activity. Primary care, ED admission and subsequent 111 call activity were most closely simulated (table 1). Primary care, ED attendance and subsequent 111 call activity were most closely simulated (figure 2B, figure 3A).

The first healthcare contact following the index 111 call in the Connected Yorkshire data was most commonly a primary care service (26 690/56 102; 47.6%). However, a large proportion (21 749/56 102; 38.8%) had no healthcare contact in the 72 hours following the index 111 call. This was mirrored by the simulation data, where the first contact following the index 111 call was a primary care service in a mean of 27 697/58 302 (47.5%) of cases, with no further healthcare contact the second most common finding (22 700/58 302; 38.9%) (figure 3B).

### 'What if?' scenario

We simulated a hypothetical scenario whereby all 111 calls with a primary care disposition received a timely primary care contact (the 'what if?' scenario). This assumed that there were no capacity constraints on the service and that all callers accessed a primary care service following the index 111 call.

This resulted in an increase in primary care contacts from a mean of 39 283 (95% CI 39 237 to 39 328) in the base simulation to 77 030 (95% CI 76 964 to 77 097); a mean percentage increase of 196.1% (95% CI 192.2% to 199.9%) (table 1). All other healthcare services saw a reduction in healthcare interactions, with the most notable reductions occurring in 999 and 111 service access (mean percentage change 78.0% and 64.2%, respectively).

### ED attendance

In the original Connected Yorkshire dataset, there were 9290 ED attendances and 1029 (11.1%) met the O'Keeffe et al[17] definition of an avoidable attendance. The simulated dataset estimated a mean ED attendance of 9337 cases (95% CI 9319 to 9356), of which, 1120 (95% CI 1114 to 1127; mean proportion, 12%, 95% CI 5.6% to 18.4%) were classed as avoidable. In the 'what if' scenario, ED attendances reduced to a mean of 8228 (95% CI 8213

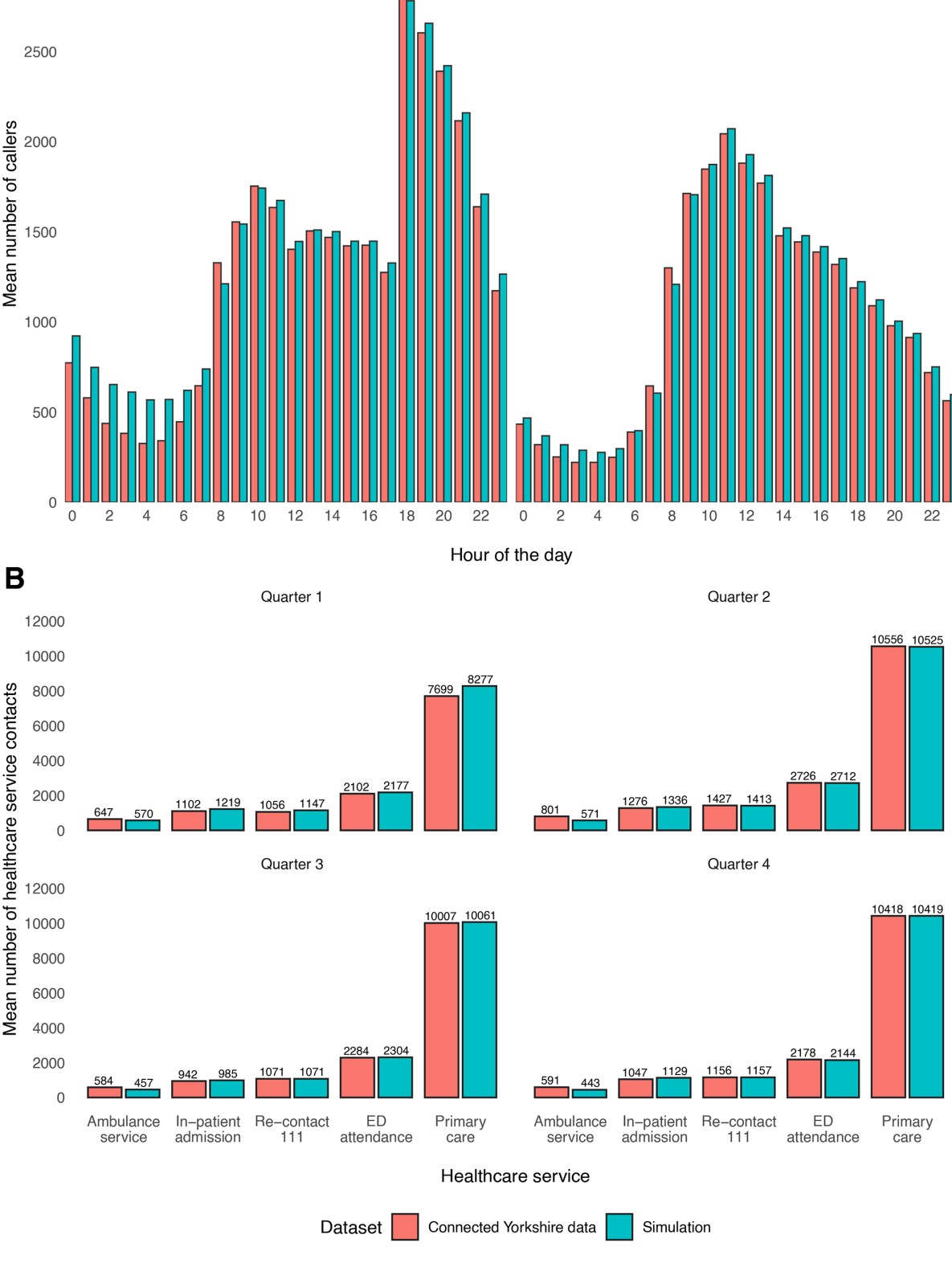

**Figure 2** (A) Comparison between connected Yorkshire and simulation data of hourly caller numbers stratified by weekday/weekend. (B) Comparison between connected Yorkshire and simulation data of healthcare service contacts, stratified by service and yearly quarter. ED, emergency department.

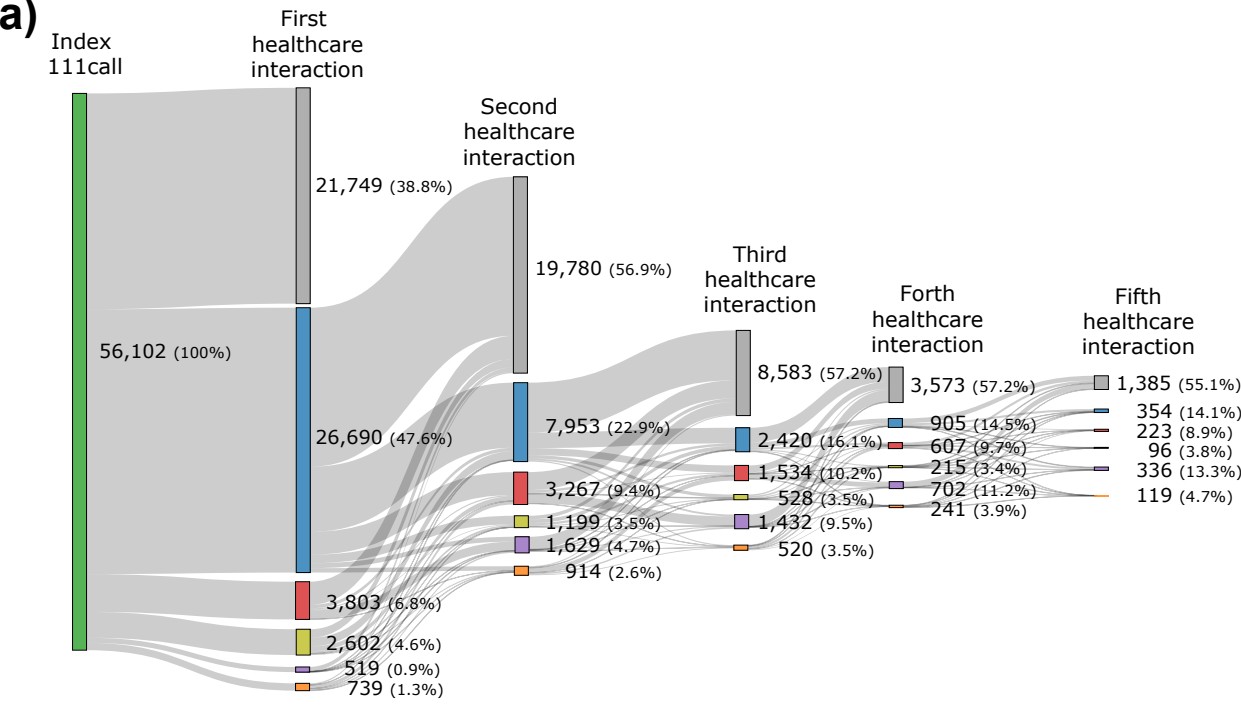

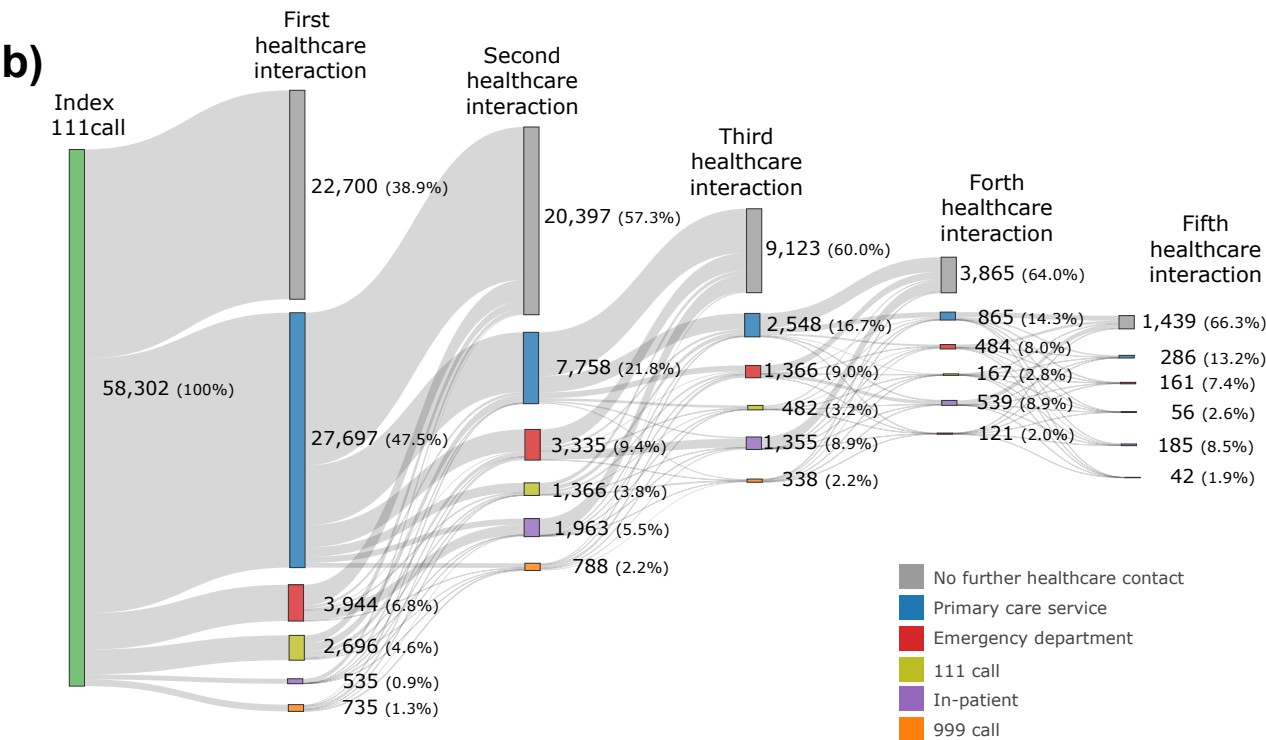

**Figure 3** (A) Sankey diagram of the connected Yorkshire data showing the first five healthcare interactions following the index 111 call. (B) Sankey diagram of the simulation showing the first five healthcare interactions following the index 111 call.

to 8244) attendances but there was little change in the proportion of avoidable attendance (mean proportion 13.3%, 95% CI 6.6% to 20%).

## DISCUSSION

Using a simple simulation model, we have been able to successfully capture the healthcare system as it relates to callers to 111 who are triaged to a primary care disposition. We used the model to estimate the impact of a hypothetical scenario in which primary care services had sufficient capacity to provide a service to all 111 callers triaged to

**Table 1** Comparison of total healthcare service contacts between the origin simulation and 'what if?' scenario

| Service | Connected Yorkshire data (N) | Simulation (mean, 95% CI) | What if? (mean, 95% CI) | Mean difference (mean 95% CI) | Mean percentage change (mean %, 95% CI) |
|---|---|---|---|---|---|
| Ambulance service | 2623 | 2042 (2032 to 2051) | 1593 (1584 to 1602) | 449 (−461 to −436) | 78 (69.8 to 86.2) |
| Emergency department | 9290 | 9337 (9319 to 9355) | 8228 (8213 to 8244) | 1109 (−1133 to −1085) | 88.1 (81.7 to 94.5) |
| Primary care | 38 680 | 39 283 (39 237 to 39 328) | 77 030 (76 964 to 77 097) | 37 748 (37 667 to 37 829) | 196.1 (192.2 to 199.9) |
| In-patient | 4367 | 4668 (4654 to 4681) | 4620 (4607 to 4632) | 48 (−66 to −30) | 99 (97 to 101) |
| 111 | 4710 | 4789 (4771 to 4806) | 3074 (3061 to 3087) | 1715 (−1737 to −1693) | 64.2 (54.7 to 73.7) |

their service and that all callers elected to contact the service. We determined that the 'base case' model sufficiently closely replicated the real-world data to be confident that its estimates of the impact of the scenario tested would be useful. While we did not monitor or simulate ambulance dispatch, we can still see the outcome of any physical attendance by an ambulance crew by monitoring ED attendances or GP contacts following a 999 call, for example.

In the simulated 'what if?' scenario, we found that ensuring a timely primary care contact for all 111 callers triaged to a primary care disposition, would lead to a significant reduction in ambulance service and 111 contacts and ED admissions. However, to achieve this, it would be necessary to almost double the number of primary care contacts provided by the current healthcare system, which, in the wider context of clinical staff shortages in general practice[20 21] and the NHS,[22] is unlikely to be feasible. While we did not conduct an economic analysis, the data suggest that the savings made by reducing certain healthcare service contacts are unlikely to offset the additional cost in providing additional primary care services.

Even if the capacity issues could be overcome, adjusting caller behaviour could be even more challenging. Almost 39% of callers in the base case model did not seek any further healthcare contact in the 72 hours after the index 111 call. Previous qualitative work on understanding why patients call an ambulance for 'primary care problems' highlighted the perceived limitations of out-of-hours primary care services by patients to address their needs and, in the case of patients with chronic conditions, previous negative experiences of community-based healthcare services being unable to 'help' would need to be addressed.[23] In addition, there are reasons why patients may not follow advice to contact primary care which are beyond the control of the health system, such as work or family/childcare commitments, transport issues (including weather-related) and demographic factors such as younger age, female sex and low socioeconomic background[24]; factors that would be unaffected by changes in service provision. That said, there is limited evidence that increasing availability of primary care services (such as offering extended hours) can lead to a

reduction in access to other services. This is particularly pertinent, since most 111 calls with a primary care disposition occur out-of-hours.[25]

This work has generated stakeholder discussion about whether the apparent supply/demand problem identified by the simulation could be addressed by using the NHS Additional Roles Reimbursement Scheme which provides funding for healthcare professionals (HCPs) to work as a multidisciplinary team within primary care to expand capacity and support delivery of new services tailored to the local population. Potential HCP roles include pharmacists, paramedics, occupational therapists and dieticians in addition to others.

In addition, they have also suggested that further qualitative work might help to understand which primary care roles best suit particular caller presentations (eg, a consultation with physiotherapist rather than a GP for a musculoskeletal presentation), which may reduce the proportion of callers who do not access healthcare services following the 111 call and are potentially left with an unmet healthcare need.

### Strengths and weaknesses

This study benefits from utilisation of a robust method (DES) that has been widely used and accepted as a method to simulate healthcare system changes.[26] In addition, the model was informed by relevant stakeholders and a linked dataset. However, it is nevertheless a simple working model, and does not take account of factors that have previously been associated with inappropriate ED attendance, such as age, sex, ethnicity and clinical input into the 111 call.[6]

The model may benefit from more extensive validation and verification than the simple 'black box' validation described here, in which outputs from the model were compared with the equivalent real-world metrics. Additional validation measures, such as performing sensitivity analyses to determine the impact of inaccuracies in modelling assumptions, could be useful in increasing confidence in the model's predictions.[27] In addition, verification measures such as extensive review of code by others can help to identify issues in the software engineering side of the development.[28] In addition, only callers who are registered with a Bradford GP were included. Given that

GP registrations are consistently higher than Office for National Statistics population estimates, it is difficult to determine what proportion of the population are not registered with a GP. Patients also have a right to opt-out of their clinical data being used for research purposes, and this comprises around 4% of patients registered with a GP in England.[29]

The data that informed the model only cover a discrete region in West Yorkshire, which may affect generalisability. Bradford is mainly an urban area and the 13th most deprived local authority in England (out of 333) based on the Index of Multiple Deprivation.[30] In addition, it is based on data collected in 2021, during the pandemic, which may not be representative of the current healthcare trajectory for patients who call 111. However, the open source nature of the model means that others can use the model to parameterise according to the data from their own regions.

Further economic and qualitative analysis would be beneficial to determine whether an intervention such as this would be cost-effective, feasible and acceptable to patients. This could potentially include ambulance attendance following 999 calls, which may have implications for ambulance service operational performance and cost-effectiveness.

## CONCLUSION

In this simulated study, ensuring timely contact with a primary care service is estimated to lead to a significant reduction in 999 and 111 calls, and ED attendances (although not avoidable ED attendance). However, this is likely to be impractical given the estimated need to almost double current primary care service provision. Further economic and qualitative research is needed to determine whether this intervention would be cost-effective and acceptable to both patients and primary care providers.

**Twitter** Richard Pilbery @999CPD

**Acknowledgements** This work uses data provided by patients and collected by the NHS as part of their care and support. The authors would also like to thank Dr. Eithne Cummins and Dr. Jon Dickson for providing a stakeholder perspective to the study and Dr. Hazel Squires for early support in operational research methodology. In addition, we are grateful for the support provided by the team at Connected Yorkshire, especially Kuldeep Sohal and John Birkinshaw.The work was supported as a project of the NIHR Applied Research Collaboration for the South West Peninsula Health Service Modelling Associates (HSMA) Programme (https://sites.google.com/nihr.ac.uk/hsma) which also provided the training required in Python and Discrete Event Simulation.

**Contributors** RP and DC conceived and designed the study. JG conducted a literature review. RP obtained the research approvals to access the datasets and acts as guarantor for the paper. RP and MS developed the code necessary to extract the data that informed the model. RP coded the model. DC provided mentoring support and guidance for the project, and delivered the training in Python and SimPy. CAO'K assisted with the design of the study. All authors drafted the manuscript and contributed substantially to its revision.

**Funding** This paper presents independent research by the NIHR Applied Research Collaboration Yorkshire and Humber (ARC YH). This work was supported by the National Institute for Health Research Applied Research Collaboration South West Peninsula and Yorkshire and Humber.

**Disclaimer** The views expressed in this publication are those of the author(s) and not necessarily those of the National Institute for Health Research or the Department of Health and Social Care.

**Competing interests** None declared.

**Patient and public involvement** Patients and/or the public were involved in the design, or conduct, or reporting, or dissemination plans of this research. Refer to the Methods section for further details.

**Patient consent for publication** Not applicable.

**Ethics approval** This study was approved by the Bradford Learning Health System Board in accordance with the Connected Yorkshire NHS Research Ethics Committee (REC) and Confidentiality Advisory Group (CAG) approvals relating to the Connected Yorkshire research database (17/EM/0254). No separate Health Research Authority (HRA) approval was required for this study.

**Provenance and peer review** Not commissioned; externally peer reviewed.

**Data availability statement** Data are available on reasonable request. The complete model, including transition probabilities and interarrival, activity and queue distributions are available from the study GitHub repository (https://github.com/RichardPilbery/MOOOD-study). The software is supplied under a GNU General Public License.

**ORCID iDs**
Richard Pilbery http://orcid.org/0000-0002-5797-9788
Daniel Chalk http://orcid.org/0000-0002-4165-4364

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
