## [Reviewer comments · BMJ Open]

ARTICLE DETAILS

TITLE (PROVISIONAL)	Modelling NHS England 111 demand for primary care services: a discrete event simulation
AUTHORS	Pilbery, Richard; Smith, Madeleine; Green, Jonathan; Chalk, Daniel; O'Keeffe, Colin

VERSION 1 – REVIEW

REVIEWER	Jorm, Louisa University of New South Wales, Centre for Big Data Research in Health
REVIEW RETURNED	29-Jun-2023

GENERAL COMMENTS	This paper presents an approach to simulating health care usage among patients triaged to primary care after a call to NHS 111, and testing the impacts of varying parameters in the simulation model to represent a hypothetical scenario whereby all patients received a contact from the primary health care service within the specified call triage time. Many aspects of the paper need to be strengthened and clarified before it is suitable for publication, not least to assist readers who are not familiar with the NHS 111 service: (1) The study objectives stated in the abstract and introduction (introduction, page 4, para 4) need to more explicitly describe what is meant by a “timely primary care service contact”. (2) It is unclear (introduction, page 4, para 2) what triage to referral to a primary care service entails. Does this mean that the patient is advised to make contact with a primary care service, or does a primary care service proactively contact the patient within a specified call triage time? (3) The methods (data, page 5, para 3) seem to suggest that healthcare system access in the 72 hours following the index call was identified by individually searching the various datasets. How was this done? Were these linked datasets? (4) The methods (data, page 5, para 3) are vague about what type of primary care contacts/services/episodes are captured – does this include calls made by the primary care service to the patient (or vice a versa) to make an appointment, or only services rendered, such as telehealth and physical visits to a service? (5) The following statement (discrete event simulation, page 7, para 5) is confusing: “Patients remain in the simulation until either they are allocated to ‘no further health care contact’ or the elapsed
---

	time exceeds 72 hours. The model runs for a period of 1 year of simulated time.” Do patients remain in the simulation for 72 hours, or for 1 year? The relationship between the conceptual model (Figure 1) and the simulation flow chart (Figure 2) is similarly confusing and needs to be clarified. (6) The methods state (analysis, page 8, para 3) that the model was evaluated by comparing quarterly aggregated health care service access by patients, and visual assessment of patient trajectory. It isn’t clear how these methods align with the findings are presented in the results. (7) The results section is very brief. As per comments (4) and (5), it isn’t clear what Table 1 (page 9) presents. What was the total number of index 111 calls? Assuming that the results presented are quarterly aggregated data, where are the results for what happened in the 72 hours after the index call? What primary care services are included? Table 1 needs to be titled and labelled clearly and the data presented in Table 1 needs to be accompanied by a textual description. The same comments apply to Table 2 (page 10). (8) The discussion (page 11, para 3) states that almost 39% of callers in the base model did not seek healthcare contact in the 72 hours after the index 111 call. Where is this finding presented in the results? (9) Where are the results re visual assessment of patient trajectories presented? (10) Because of the lack of clarity about what primary care service exactly are being counted, it unclear what doubling the number of primary health care “contacts” (discussion, page 11, para 2) under the “what if” scenario actually means. (11) The current discussion section needs to be thoroughly checked and revised after satisfactorily addressing issues (1)-(10) above.
--	---

REVIEWER	Todd, Verity Auckland University of Technology, Paramedicine
REVIEW RETURNED	04-Jul-2023

GENERAL COMMENTS	The authors present a study evaluating the impact of improved access to primary care on ambulance usage, 999 call volume and ED presentations. The authors used a year of real world data from the Yorkshire region to generate simulation data for the improved access model for callers triaged to primary care. The study found that there would be a notable decrease in 999 call volume and ED attendances, but that to achieve this reduction a doubly in the access to primary healthcare in a timely manner would be required. The research is clearly and concisely presented and addresses a pertinent question around supporting the low acuity workload. However, I am not sure how broadly applicable the findings are based on the relatively small region investigated. Abstract: 1. Why is ambulance use not included in the numbers presented in the results? Is ambulance use looked at, or only the volume of 999 calls?
---

2. I think the percentage changes are easier to interpret for your results and should also be included.

Introduction:

3. Does the 999 service offer access to telehealth providers?

Methods:

4. How well captured is enrolment with a GP within the area? For example, in our locality, there are discrepancies between ethnicity groups, such that our most at-risk groups are underrepresented in enrolment with primary healthcare providers.

5. Could you please explain how patient data was linked for each step of the healthcare journey? Was analysis restricted to only those patients for whom a unique patient identifier number was available?

6. What is the geographic spread of the Bradford region? Is there a mix of both urban and rural residents?

7. "Timely" has not been specifically described in the manuscript. What is the specified call triage time?

8. Please replace the abbreviation PPI with the full term.

Results:

9. Please provide figure legends for the supplementary figures.

10. Table 1: Consider either replacing the column heading "cYorkshire2021" with something easier to follow, e.g. Yorkshire data or real data, or defining the abbreviation.

11. Table 2: Shouldn't the percentage for Primary Care be 196%, reflecting the almost doubling of resource required here?

12. Table 1 and Table 2: I suggest that these tables are combined, such that the cYorkshire2021 data is included as the first column in Table 2, as the Simulation data is repeated across both tables.

Discussion/Conclusion/Limitation:

13. Could you please expand on the Additional Roles Reimbursement scheme – how does this relate to other jurisdictions outside of the UK?

14. I have concerns around the applicability of the study outside of both this region within the UK, and to other jurisdictions – points that you have raised within your Discussion.

References:

15. A large number of websites and text books are referenced within this manuscript – much more than would usually be seen within a journal publication. Perhaps this is a reflection of the in silico field (e.g. refs 7-10)? Are there alternative peer-reviewed citations that could be used instead?

VERSION 1 – AUTHOR RESPONSE

Comment	Response	Location/notes
The study objectives stated in the abstract and introduction (introduction, page 4, para 4) need to more explicitly describe what is meant by a “timely primary care service contact”.	Definition of timely primary care contact provided in the introduction	Page 4, second paragraph
It is unclear (introduction, page 4, para 2) what triage to referral to a primary care service entails. Does this mean that the patient is advised to make contact with a primary care service, or does a primary care service proactively contact the patient within a specified call triage time?	Clarification provided. Triage dispositions include a mixture of callers being asked to contact a primary care service themselves, and referrals being made by 111 for make contact with the caller	Page 4, second paragraph
The methods (data, page 5, para 3) seem to suggest that healthcare system access in the 72 hours following the index call was identified by individually searching the various datasets. How was this done? Were these linked datasets?	Patients in the Connected Yorkshire datasets are allocated a unique identifier which consistently identifies the patient across all datasets. Text adjusted to make this clearer.	Page 5, Data sub-section.
The methods (data, page 5, para 3) are vague about what type of primary care contacts/services/episodes are captured – does this include calls made by the primary care service to the patient (or vice a versa) to make an appointment, or only services rendered, such as telehealth and physical visits to a service?	It includes both, but also other services other than a primary care physician (GP). Text updated to clarify this.	Page 5-6, Data sub-section
The following statement (discrete event simulation, page 7, para 5) is confusing: “Patients remain in the simulation until either they are allocated to ‘no further health care contact’ or the elapsed time exceeds 72 hours. The model runs for a period of 1 year of simulated time.” Do patients remain in the simulation for 72 hours, or for 1 year? The relationship between the conceptual model (Figure 1) and the simulation flow chart (Figure 2) is similarly confusing and needs to be clarified.	It’s the former. Patients remain for a maximum of 72 hours, but the model generates a year’s worth of calls. Text adjusted to make this clearer. We’ve revised Figure 1 and 2 to make the link	Page 8, first two paragraphs

	between the models clearer. The healthcare system is complicated, not least because even within 72 hours, callers can transit between several different services (including returning to the same service). Programmatically, processes for a single caller are conducted in a sequential manner, but operate within a loop, allowing for multiple accesses to the various services. In other words, a single caller's journey in the model is a series of interactions with services (including interacting with the same service again) determined by probabilities drawn from the real-world data.	
The methods state (analysis, page 8, para 3) that the model was evaluated by comparing quarterly aggregated health care service access by patients, and visual assessment of patient trajectory. It isn't clear how these methods align with the findings are presented in the results.	Text updated and additional figures used to undertake the assessment have been included (Figure 3 and the Sankey diagrams found in Supplementary 1 and 2)	Figure 3, Supplementary material
The results section is very brief. As per comments (4) and (5), it isn't clear what Table 1 (page 9) presents. What was the total number of index 111 calls? Assuming that the results presented are quarterly aggregated data, where are the results for what happened in the 72 hours after the index call? What primary care services are included? Table 1 needs to be titled and labelled clearly and the data presented in Table 1 needs to be accompanied by a textual description. The same comments apply to Table 2 (page 10).	Result section expanded with additional description. Table 1 (including its caption) revised and textual explanation provided. Tables 1 and 2 have now been combined. The quarterly aggregate data was originally in the supplemental materials, but has now been incorporated into Figure 3. The sankey diagrams tracking patient healthcare trajectory over the 72 hours following the index call have been added as supplemental	Results section, Table 1, 2, Figure 3, Supplementary 1 and 2

	material.	
The discussion (page 11, para 3) states that almost 39% of callers in the base model did not seek healthcare contact in the 72 hours after the index 111 call. Where is this finding presented in the results?	Results section revised to include this data.	Results section and Supplementary 2
Where are the results re visual assessment of patient trajectories presented?	These have now been added and can be found within Figure 3 and Supplementary 1 and 2.	
Because of the lack of clarity about what primary care service exactly are being counted, it unclear what doubling the number of primary health care “contacts” (discussion, page 11, para 2) under the “what if” scenario actually means.	Additional detail around what a ‘primary care service’ has been provided.	Page 4, second paragraph
The current discussion section needs to be thoroughly checked and revised after satisfactorily addressing issues (1)-(10) above.	Checked and revised as required.	Discussion section
Why is ambulance use not included in the numbers presented in the results? Is ambulance use looked at, or only the volume of 999 calls?	No, ambulance dispatch was not included in this model. We aimed to keep the model as simple as possible. However, since we tracked the caller’s healthcare trajectory, we can still see the outcome of any physical attendance by an ambulance crew, for example an ED attendance, or GP contact. This would be an interesting extension, particularly with respect to ambulance service operational performance and cost effectiveness, but was outside the scope of this feasibility.	Acknowledgement of this in the first paragraph of the discussion section, page 11–12.
I think the percentage changes are easier to interpret for your results and should also be included.	Table 1 updated with percent changes rather than difference in proportions.	Table 1

Intro: Does the 999 service offer access to telehealth providers?	No. There are some clinicians working within the emergency operations centre, but during the course of this study (up to the present), they are more focussed on scanning the job stack for cases that can either be downgraded and signposted to alternative services, or upgraded, because of the excessive time that has elapsed e.g. elderly fall patients with no injury, for example.	
Methods: How well captured is enrolment with a GP within the area? For example, in our locality, there are discrepancies between ethnicity groups, such that our most at-risk groups are underrepresented in enrolment with primary healthcare providers.	This is difficult to answer, since the number of GP registrations consistently exceeds the population estimate i.e. more patients are registered with a GP than are actually estimated to be present in the population! The NHS does have a data opt-out, whereby patients can refuse to allow their data to be used for research purposes. This comprises around 4% of patients registered with GP practices in the Connected Yorkshire catchment area. Limitations updated to acknowledge these potential issues.	Strengths and weaknesses
Methods: Could you please explain how patient data was linked for each step of the healthcare journey? Was analysis restricted to only those patients for whom a unique patient identifier number was available?	Correct. Submissions made by the various health services only include those with an NHS number. This is pretty ubiquitous in most in-hospital services. Even for 111, NHS numbers are available for around 98% of callers.	Page 5

	Data sub-section expanded to clarify this.	
Methods: What is the geographic spread of the Bradford region? Is there a mix of both urban and rural residents?	It is mostly urban and deprived. This is addressed in the strengths and weaknesses section.	Strengths and weaknesses
Methods: "Timely" has not been specifically described in the manuscript. What is the specified call triage time?	Definition of timely primary care contact provided in the introduction	Page 4, second paragraph
Methods: Please replace the abbreviation PPI with the full term.	Heading replaced	Page 7
Results: Please provide figure legends for the supplementary figures.	Figure legends added	
Results: Table 1: Consider either replacing the column heading "cYorkshire2021" with something easier to follow, e.g. Yorkshire data or real data, or defining the abbreviation.	Column heading updated	Table 1
Results: Table 2: Shouldn't the percentage for Primary Care be 196%, reflecting the almost doubling of resource required here?	We had originally calculated the proportion difference as opposed to percentage change. We have now revised Table 1 (previously Table 2) so that the final column is the expected percentage change, which as the other reviewer has pointed out, is easier to interpret.	Table 1
Results: Table 1 and Table 2: I suggest that these tables are combined, such that the cYorkshire2021 data is included as the first column in Table 2, as the	Agreed. Tables combined into Table 1.	Table 1

Simulation data is repeated across both tables.		
Discussion: Could you please expand on the Additional Roles Reimbursement scheme – how does this relate to other jurisdictions outside of the UK?	Brief explanation given. As far as we are aware, this is an NHS specific initiative, although funding for additional healthcare professional roles to work with doctors in primary care could potentially be possible elsewhere.	Discussion, page 13.
Discussion: I have concerns around the applicability of the study outside of both this region within the UK, and to other jurisdictions – points that you have raised within your Discussion.	We agree, which is why we highlighted this as a study weakness. However, the practice of using modelling and simulation to understand a complex health system, particularly prior to changes to the system are to be made could of benefit to stakeholders and decision makers. Providing an applied example in the published literature is of benefit, in our opinion.	
References: A large number of websites and text books are referenced within this manuscript – much more than would usually be seen within a journal publication. Perhaps this is a reflection of the in silico field (e.g. refs 7-10)? Are there alternative peer-reviewed citations that could be used instead?	References 7–10 updated with alternative peer-reviewed citations.	

VERSION 2 – REVIEW

REVIEWER	Todd, Verity Auckland University of Technology, Paramedicine
REVIEW RETURNED	11-Aug-2023

GENERAL COMMENTS	I am satisfied that the authors have addressed the comments raised in the initial review. The following minor comments should be considered: Supplementary Information – in the Sankey diagrams, it would be useful to have percentages as well as number of events. There is a reference missing from this sentence: Patients also have a right to opt-out of their clinical data being used for research purposes, and this comprises around 4% of patients registered with a GP in England [REF]. I don't think that Figure 1 adds to the manuscript, and makes the process challenging to follow. The Sankey diagrams give a better idea of the complexity that can arise from these patient interactions. I would recommend that a Sankey diagram be elevated into the main manuscript, and Figure 1 is removed.
---

VERSION 2 – AUTHOR RESPONSE

Comment	Response	Location/notes
Please include your study location in your title for the sake of international readers	Country added to title	Title
Supplementary Information – in the Sankey diagrams, it would be useful to have percentages as well as number of events.	Figures updated with percentages	Figure 3
There is a reference missing from this sentence: Patients also have a right to opt-out of their clinical data being used for research purposes, and this comprises around 4% of patients registered with a GP in England [REF].	Apologies, this was a relic in the marked up copy. The actual manuscript did include the reference (29).	References
I don't think that Figure 1 adds to the manuscript, and makes the process challenging to follow. The Sankey diagrams give a better idea of the complexity that can arise from these patient interactions. I would recommend that a Sankey diagram be elevated into the main manuscript, and Figure 1 is removed.	Figure 1 removed. Supplementary material elevated into main manuscript	Figure 3